# Diabetes increases the risk of COVID-19 in an altitude dependent manner: An analysis of 1,280,806 Mexican patients

Juan Alonso Leon-Abarca[1,2], Arianna Portmann-Baracco[1,2], Mayte Bryce-Alberti[1,2], Carlos Ruiz-Sánchez[1,2], Roberto Alfonso Accinelli[1,2,3¤]*, Jorge Soliz[4,5], Gustavo Francisco Gonzales[1,6]

**1** Instituto de Investigaciones de la Altura, Universidad Peruana Cayetano Heredia, Lima, Perú, **2** Facultad de Medicina Albero Hurtado, Universidad Peruana Cayetano Heredia, Lima, Perú, **3** Servicio de Neumología, Hospital Cayetano Heredia, Lima, Perú, **4** Institute Universitaire de Cardiologie et de Pneumologie de Québec [IUCPQ], Faculty of Medicine, Université Laval, Québec, QC, Canada, **5** High Altitude Pulmonary and Pathology Institute (HAPPI-IPPA), La Paz, Bolivia, **6** Laboratorios de Investigación y Desarrollo [LID], Facultad de Ciencias y Filosofía, Alberto Cazorla Tálleri, Universidad Peruana Cayetano Heredia, Lima, Perú

¤ Current address: Av. Honorio Delgado, San Martín de Porres, Lima, Perú
* roberto.accinelli@upch.pe

**Data Availability Statement:** The data used in the current paper is freely available at the Mexican government's COVID-19 open data repository:

## Abstract

### Aims

The objective of this study is to analyze how the impact of Diabetes Mellitus [DM] in patients with COVID-19 varies according to altitudinal gradient.

### Methods

We obtained 1,280,806 records from adult patients with COVID-19 and DM to analyze the probability of COVID-19, development of COVID-19 pneumonia, hospitalization, intubation, admission to the Intensive Care Unit [ICU] and case-fatality rates [CFR]. Variables were controlled by age, sex and altitude of residence to calculate adjusted prevalence and prevalence ratios.

### Results

Patients with DM had a 21.8% higher prevalence of COVID-19 and an additional 120.2% higher prevalence of COVID-19 pneumonia. The adjusted prevalence was also higher for these outcomes as well as for hospitalization, intubation and ICU admission. COVID-19 and pneumonia patients with DM had a 97.0% and 19.4% higher CFR, respectively. With increasing altitudes, the probability of being a confirmed COVID-19 case and the development of pneumonia decreased along CFR for patients with and without DM. However, COVID-19 patients with DM were more likely to require intubation when residing at high altitude.

https://www.gob.mx/salud/documentos/datos-abiertos-152127.

**Funding:** The authors received no specific funding for this work.

**Competing interests:** The authors declare that no competing interests exist.

## Conclusions

The study suggests that patients with DM have a higher probability of being a confirmed COVID-19 case and developing pneumonia. Higher altitude had a protective relationship against SARS-CoV-2 infection; however, it may be associated with more severe cases in patients with and without DM. High altitude decreases CFR for all COVID-19 patients. Our work also shows that women are less affected than men regardless of altitude.

## Introduction

Among the most frequent comorbidities in patients with COVID-19 is Diabetes Mellitus [DM]. DM is a risk factor for SARS-CoV-2 virus infection, as it increases disease severity and mortality [1]. Similar associations exist between DM and morbidity in past viral pandemics, such as in SARS of 2002, influenza A [H1N1] of 2009, and Middle East Respiratory Syndrome [MERS] of 2012 [2–4].

The predisposition that patients with DM have to develop severe cases of COVID-19 can be explained by the deregulation of the Renin-Angiotensin-Aldosterone System [RAAS], the deterioration of the inflammatory response, the hypercoagulable state, and the physiological and structural lung abnormalities caused by hyperglycemia. Poor glucose control in patients with DM could promote glycosylation of angiotensin-converting enzyme 2 [ACE 2], the gateway for SARS-CoV-2 in the host [5].

Remarkably, the prevalence and severity of COVID-19 [and other respiratory viral diseases] decreases with the altitudinal gradient [6–9]. Lung expression of ACE2 appears to decrease in chronic hypoxic conditions found in high-altitude settings, which may explain why highland dwellers appear dramatically less affected by the pandemic compared to lowland dwellers [6, 10, 11]. Additionally, reports showed that hypoxic conditions improved glycemic homeostasis in patients with DM [6]. However, to our knowledge, there are currently no studies simultaneously evaluating the impact of DM and altitude on COVID-19. Studies that compare the impact of sex on these diseases are also scarce. As such, we analyzed the dossier of 1,280,806 adult subjects [men and women] provided by the Mexican government through an open database [12]. Mexico is globally the fourth country most affected by COVID-19, surpassed only by the death tolls of the USA, Brazil and India. Additionally, it has ranked as the country with most health-care workers COVID-19 mortality. In spite of this, Mexico maintains a strategy of monitoring hospital capacity and not testing widely or contact tracing [13].

## Research design and methods

Our study included the records of adult (between 20 and 90 years) Mexican patients who had symptoms compatible with COVID-19 in the last 7 days as described by the national Mexican guidelines. Governmental procedure states that patients in respiratory distress must be tested through a nasopharyngeal swab for a SARS-CoV-2 RT-PCR assay. Through sentinel surveillance at pre-designed healthcare centers across the country (475 Viral respiratory disease monitoring units, USMER in Spanish), one out of ten patients with suspected mild COVID-19 (ambulatory, without hospitalization) and every patient who required hospitalization due to COVID-19 suspicion were tested for SARS-CoV-2 infection with a nasopharyngeal swab. Before testing, patients were asked about pre-existing COVID-19 risk factors such as DM, obesity, arterial hypertension, chronic obstructive pulmonary disease, asthma, smoking,

immunodeficiency, cardiovascular diseases, chronic kidney disease and pregnancy. An open repository by the Mexican government provided the data used in this study [12]; therefore, there were constraints for the extent of variables used. As eligible registers up to September 9[th] 2020 were used (Accession date);1280806 patients with COVID-19 were included in the study after excluding patients with missing DM status records, undefined RT-PCR results and unknown place of residence. Patients with additional risk factors were not excluded. We defined a COVID-19 case as a patient who tested positive for a SARS-CoV-2 RT-PCR assay and a COVID-19 pneumonia as the subset of patients who also had pneumonia, either by clinical or radiologic diagnosis. Altitude was calculated as the average altitude of the municipalities listed in the patients' records and then the data was organized into groups corresponding to altitudes below 1500 m and above 1500 m. Considering data source availability, we defined high altitude from 1500 m up to records of patients who lived at 3500 meters above sea level [14]. We then sought to analyze patients with DM with a definite RT-PCR result to evaluate whether the individual probability of SARS-CoV-2 virus infection, development of COVID-19 pneumonia, hospitalization, intubation, Intensive Care Unit [ICU] admission and mortality varied according to age and current altitude of residence.

Poisson regressions with robust standard errors were applied to relate sex, altitude and DM status to each of the four COVID-19 and COVID-19 pneumonia outcomes: hospitalization, intubation, ICU admission and death [case-fatality rate or CFR]. We considered interaction between variables for DM + age, DM + sex and DM + altitude to account for the possible cluster effect of patients around particular healthcare centers in diverse regions. Post-estimation marginal analysis was performed to find the adjusted prevalence [aP] and adjusted prevalence ratios [aPR] of the independent variables. Continuous variables were employed to construct binomial regression models to produce predicted probability plots of each outcome to display the interactions between variables as probability trends along 95% confidence bands for ages between 20–90 years and altitudes up to 3500m. For data analysis, the STATA 14.0 program was used considering a p value of 0.05 as the statistically significant threshold. The study received approval from Universidad Peruana Cayetano Heredia's ethics committee with registration number SIDISI 202908.

## Results

We present summary characteristics of the study population in **Table 1**. Out of the total number of records, 12.97% (95% CI: 12.92–13.03%) had DM. The mean age was 43.8 years (SD: 15.4 years) and the group with DM was, on average, 15.6 years older than the group without DM.

Females had a slight predominance with an overall male-to-female ratio of 0.968. After symptom onset, the group with DM had a tendency to attend a healthcare center a day later than the group without DM, and once admitted, died a day sooner than those without DM (p<0.0001).

All patients with DM had one or two additional risk factors, while 65.56% of patients without DM had no additional risk factors (p<0.0001). Patients with DM, compared to those without DM, had a 21.8% higher rate of COVID-19, threefold rates of COVID-19 pneumonia, and were more likely to be hospitalized, intubated, admitted to the ICU, and have a fatal outcome. After adjusting for sex, age and altitude, patients with DM displayed a higher probability of COVID-19 (+21.8%), COVID-19 pneumonia (+120.2%), hospitalization (+85.9%), intubation (+11.1%), ICU admission (+7.2%), and CFR (+97.0%), compared to those without DM (p<0.0001). Patients with DM that developed COVID-19 pneumonia had a higher probability

**Table 1. Demographic and clinical characteristics of the study population.**

| Characteristic | Patients with DM (N = 166,167) | Patients without DM (N = 1'114,639) | Total (N = 1'280,806) |
|---|---|---|---|
| Mean Age (SD)–yr. | 57.4 (13.40) | 41.8 (14.70) | 43.8 (15.40) |
| Male Sex–no. (%) | 84,723 (50.99) | 545,270 (48.92) | 629,993 (49.19) |
| Female Sex–no. (%) | 81,444 (49.01) | 569,369 (51.08) | 650,813 (50.81) |
| Median (IQR) days of duration of symptoms before healthcare consult | 4 (4) | 3 (4) | 3 (4) |
| Median (IQR) days from admission to death by COVID-19 | 5 (9) | 6 (9) | 6 (8) |
| Risk factors–no./total no. (%) | | | |
| 0 | 0 (0.00) | 730,703 (65.56) | 730,703 (57.05) |
| 1 | 47,741 (28.73) | 285,266 (25.59) | 333,007 (26.00) |
| 2 | 118,426 (71.27) | 98,670 (8.85) | 217,096 (16.95) |
| Altitude of residence—no./total no. (%) | | | |
| 0–1000 meters | 66,857 (40.23) | 393,204 (35.28) | 460,061 (35.92) |
| 1000–2000 meters | 38,234 (23.01) | 267,836 (24.03) | 306,070 (23.90) |
| 2000–3000 meters | 60,986 (36.70) | 453,599 (40.69) | 514,585 (40.18) |
| Outcome for COVID-19 disease–no. (%) | | | |
| Cases | 99,479 (59.87) | 515,094 (46.21) | 614,573 (47.98) |
| Admission | 51,429 (51.70) | 104,677 (20.32) | 156,106 (25.40) |
| Intubation | 6,734 (6.77) | 12,218 (2.37) | 18,952 (3.08) |
| ICU admission† | 4,494 (4.52) | 8,596 (1.67) | 13,090 (2.13) |
| Death | 25,952 (26.09) | 41,464 (8.05) | 67,416 (10.97) |
| Outcome for COVID-19 pneumonia–no. (%) | | | |
| Cases | 38,847 (23.38) | 81,316 (7.30) | 120,163 (9.38) |
| Admission | 35,244 (90.73) | 67,233 (82.68) | 102,477 (85.28) |
| Intubation | 5,717 (14.72) | 10,064 (12.38) | 15,781 (13.13) |
| ICU admission†† | 3,992 (10.28) | 7,354 (9.04) | 11,346 (9.44) |
| Death | 19,333 (49.77) | 30,750 (37.82) | 50,083 (41.68) |

Percentages expressed as total number of observations of the group categorized by DM diagnostic and the rest of variables. Clinical outcomes express the % of patients within each group observed of the total number of cases by DM status. All comparisons between patients with DM and without DBM stand at p<0.0001, except † = 0.005 and †† = 0.062

of hospitalization (+8.0%), intubation (+8.1%), ICU admission (+4.5%), and CFR (+19.4%), compared to those without DM (p<0.0001) (Table 2).

We present an analysis of patients by DM status, age and the altitude of residence cut-off at 1500 m in Fig 1. The probability of contracting COVID-19 in patients with and without DM increased with age and was higher for patients living below 1500 m compared to that of those living above 1500 m (p<0.05). However, these variables were affected to a lesser magnitude in patients without DM. Patients aged ≥55 years without DM who lived below 1500 m had a higher risk of contracting COVID-19 compared to patients with DM living above 1500m (p<0.05) (Fig 1A). Once COVID-19 ensued, patients with DM had a higher probability of developing pneumonia than patients without DM p<0.05 and there were no differences between those living below or above 1500 m (p≥0.05) (Fig 1B). The probability of admission for patients with COVID-19 and COVID-19 pneumonia increased with age and was higher for patients with DM who lived below 1500 m compared to those who lived above 1500 m (p<0.05) (Fig 1C and 1D). The case-fatality rate increased with age and was higher for patients living below 1500 m compared to that of those living above 1500 m (p<0.05). Patients at all

**Table 2. Adjusted prevalences (aP) by age, sex and altitude.**

| Outcomes | n | All patients | | Patients with DM | | Patients without DM | |
|---|---|---|---|---|---|---|---|
| | | aPR | 95% CI | aP | 95% CI | aP | 95% CI |
| **COVID-19** | 1'280,806 | 1.218 | 1.21–1.23 | 57.3% | 57.0–57.6 | 47.0% | 46.9–47.1 |
| **Admission** | 614,573 | 1.859 | 1.84–1.88 | 41.9% | 41.5–42.2 | 22.5% | 22.4–22.6 |
| **Intubation** | 155,984 | 1.111 | 1.08–1.14 | 13.1% | 12.8–13.4 | 11.8% | 11.6–12.0 |
| **ICU admission** | 155,984 | 1.072 | 1.03–1.11 | 8.8% | 8.6–9.1 | 8.2% | 8.1–8.4 |
| **Death/CFR** | 614,573 | 1.970 | 1.94–2.0 | 18.3% | 9.2–9.4 | 9.3% | 18.1–18.6 |
| **COVID-19 pneumonia** | 1'280,787 | 2.202 | 2.17–2.23 | 17.9% | 17.7–18.1 | 8.1% | 8.1–8.2 |
| **Admission** | 120,163 | 1.080 | 1.07–1.08 | 90.3% | 90.0–90.6 | 83.6% | 83.4–83.9 |
| **Intubation** | 102,422 | 1.081 | 1.05–1.11 | 16.3% | 15.9–16.7 | 15.1% | 14.8–15.3 |
| **ICU admission** | 102,422 | 1.045 | 1.01–1.08 | 11.4% | 11.1–11.8 | 10.9% | 10.7–11.2 |
| **Death/CFR** | 120,163 | 1.194 | 1.18–1.21 | 47.1% | 46.6–47.6 | 39.4% | 39.1–39.7 |

Adjusted prevalence ratios (aPR) and adjusted prevalences (aP) done by age, sex and altitude for patients with and without DM shown along 95% CIs. All results are statistically significant at p<0.05.

altitudes with DM had a higher CFR compared to those without DM (p<0.05) (**Fig 1E and 1F**).

The probability of being a confirmed case of COVID-19 decreased by 11.3% in patients with DM and by 30% in patients without DM at 3500m compared to that of sea level (**Fig 2A**). The altitude effect influenced less sharply the probability of developing COVID-19 pneumonia with decreasing trends for all patients (**Fig 2B**). The probability of admission for all COVID-19 and COVID-19 pneumonia patients decreased slightly with higher altitudes for patients with DM (p<0.05), while it increased for patients without DM (p<0.05) (**Fig 2C and 2D**). The CFR decreased with higher altitudes. (**Fig 2E and 2F**). Additionally, males with and without DM at all altitudes had a higher risk of all these outcomes compared to female patients (p<0.05) (**Fig 2A–2F**).

The overall probability of intubation increased with age. Patients with DM who lived above 1500 m had a higher probability of intubation compared to those who lived below 1500 m p<0.05. This probability was higher for patients with DM compared to that of patients without DM up to 60 (at <1500 m) (p<0.05) and 65 years old (at >1500 m) (p<0.05) (**Fig 3A and 3B**).

According to altitude, the probability of intubation in patients with DM increased by up to 33.3% in males and 37.5% in females at 3500 m compared to that of sea level. This probability increased with lower magnitudes for patients without DM. Patients with DM had a higher risk of intubation than patients without DM (p<0.05), except for male patients living below 1000m (p≥0.05) (**Fig 3C and 3D**). The probability of intubation by age, sex and altitude of residence in patients with COVID-19 pneumonia is shown in **Fig 4**.

The overall probability of ICU admission decreased with age for patients with DM and increased for patients without DM. Patients with DM had a higher probability of this event compared to patients without DM up to 45 (<1500 m) (p<0.05) and 60 years old (>1500 m) (p<0.05). The mean probability of ICU admission was lower for patients who lived above 1500 m compared to those who lived below 1500 m (p<0.05) (**Fig 3E and 3F**). The probability of ICU admission decreased with altitude for all patients, however, no significant differences were found between men and women with or without DM (p≥0.05). The mean probability of ICU admission seemed to decrease with lower magnitudes for patients with DM compared to that of patients without DM; however, no significant differences were found (p≥0.05) (**Fig 3G**

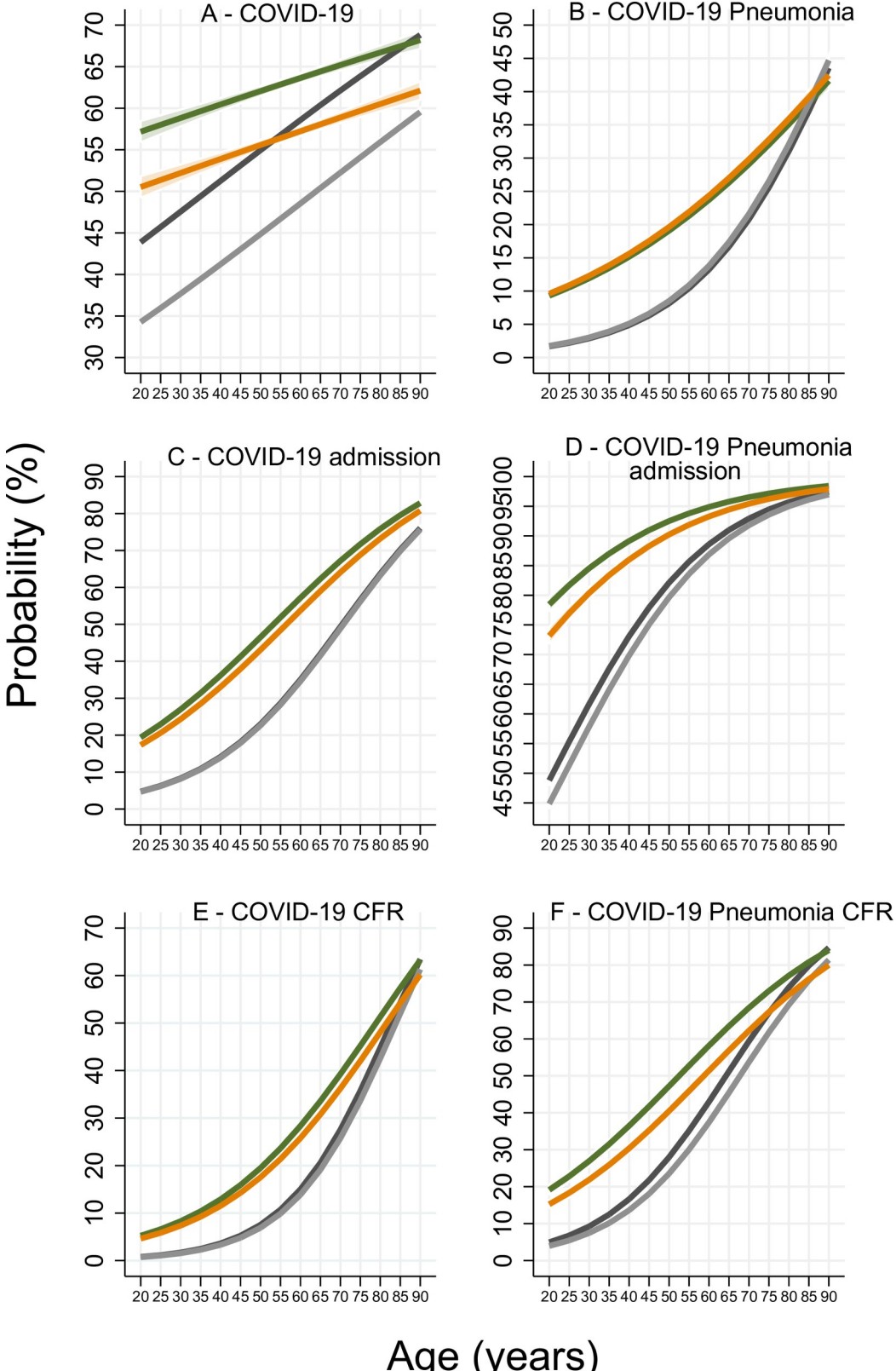

Green: DM<1500m Orange: DM>1500m. Dark gray: No DM<1500m Light gray: No DM>1500m

**Fig 1. Event probability by age.** Panel A shows that the probability of contracting COVID-19 increased with age and was higher for patients who lived below 1500 m [p<0.05]. Panel B shows that there were no significant differences is developing pneumonia between those living below or above 1500 m [p≥0.05]. Panel C-D show that the probability of admission or COVID-19 and COVID-19 pneumonia is higher for patients with DM who lived below 1500 m [p<0.05]. Panel E-F show that case fatality rate increased with age and was higher for patients with DM and for patients living below 1500 m [p<0.05].

**and 3H**). The probability of ICU admission by age, sex and altitude of residence in patients with COVID-19 pneumonia is shown in **Fig 4**.

## Discussion

Our analysis reveals that patients with DM had a 22.8% higher rate of COVID-19 and a three-fold rate of developing COVID-19 pneumonia (**Table 1**). DM is considered a risk factor for COVID-19 since it is one of the most frequent comorbidities found in these patients [15]. Two large meta-analyses with a total of 46,248 and 76,993 COVID-19 patients established a DM prevalence of 8.6% and 7.9% respectively [16, 17].

After adjusting by altitude, age, and sex, showed that the Mexican group of patients with DM was more prone to hospitalization, intubation, ICU admission, and death compared to the group without DM (**Table 2**). These results agree with a meta-analysis that includes 6452 patients from 30 studies and shows that DM is associated with higher mortality (RR 2.38 [1.88, 3.03], p < 0.001), severe COVID-19 (RR 2.45 [1.79, 3.35], p < 0.001), ARDS (RR 4.64 [1.86, 11.58], p = 0.001) and disease progression (RR 3.31 [1.08, 10.14], p = 0.04) [18]. Other reports present an association between pre-existence of DM and COVID-19 severity. These studies, which included 173 and 140 severe cases of COVID-19, report that 16.2% and 12% of patients had DM [19, 20]. The biochemical profile of patients with and without DM demonstrated that patients with DM had higher values of inflammatory markers, tissue enzymes, leukocyte counts and D-dimer. This group also presented lower numbers of lymphocytes and albumin [21]. For patients with a fatal outcome, DM prevalence rises up to 20–30%. Even without other comorbidities, patients with DM have a higher mortality than patients without DM [15]. Nevertheless, other comorbidities can influence COVID-19 progression in patients with DM. An analysis of several risk factors showed that early-onset DM increases the risk of hospitalization and that obesity mediates 49.5% of the effect of DM on COVID-19 lethality [22].

In our study, the group with DM attended healthcare centers a day later after symptom onset than the group without DM (**Table 1**). This could explain the higher mortality among COVID-19 patients with DM. Classical COVID-19 milder symptoms occur more frequently in patients with DM, thereby challenging and delaying diagnosis [15].

Our study aligns with the premise that patients with DM are more susceptible to infections caused by respiratory viruses. As such, during the SARS outbreak in 2002, DM acted as an independent risk factor for complications and mortality [2]. Similarly, the presence of DM tripled the risk of hospitalization and quadrupled the risk of ICU admission during the 2009 influenza A [H1N1] outbreak [3]. During the 2012 MERS outbreak, DM was prevalent in 50% of cases and the OR for severe or critical MERS-CoV was 7.2–15.7 compared to the total population [4]. Furthermore, there is a 22% population-attributable risk percentage for mortality related to influenza and pneumonia in patients with DM compared to patients without DM [23]. During the 1957–1960 and the 1986 influenza epidemics, DM led to a higher mortality as well as a prolonged duration of symptoms [24]. More recently, Beumer et al. reported that amongst hospitalized patients with influenza, mortality was more frequent in patients with DM [OR 7.09, CI 95% 1.23–49.75; p = 0.04] [25]. The proposed mechanisms for these outcomes include glycemic fluctuations that cause pulmonary endothelial dysfunction, elevated

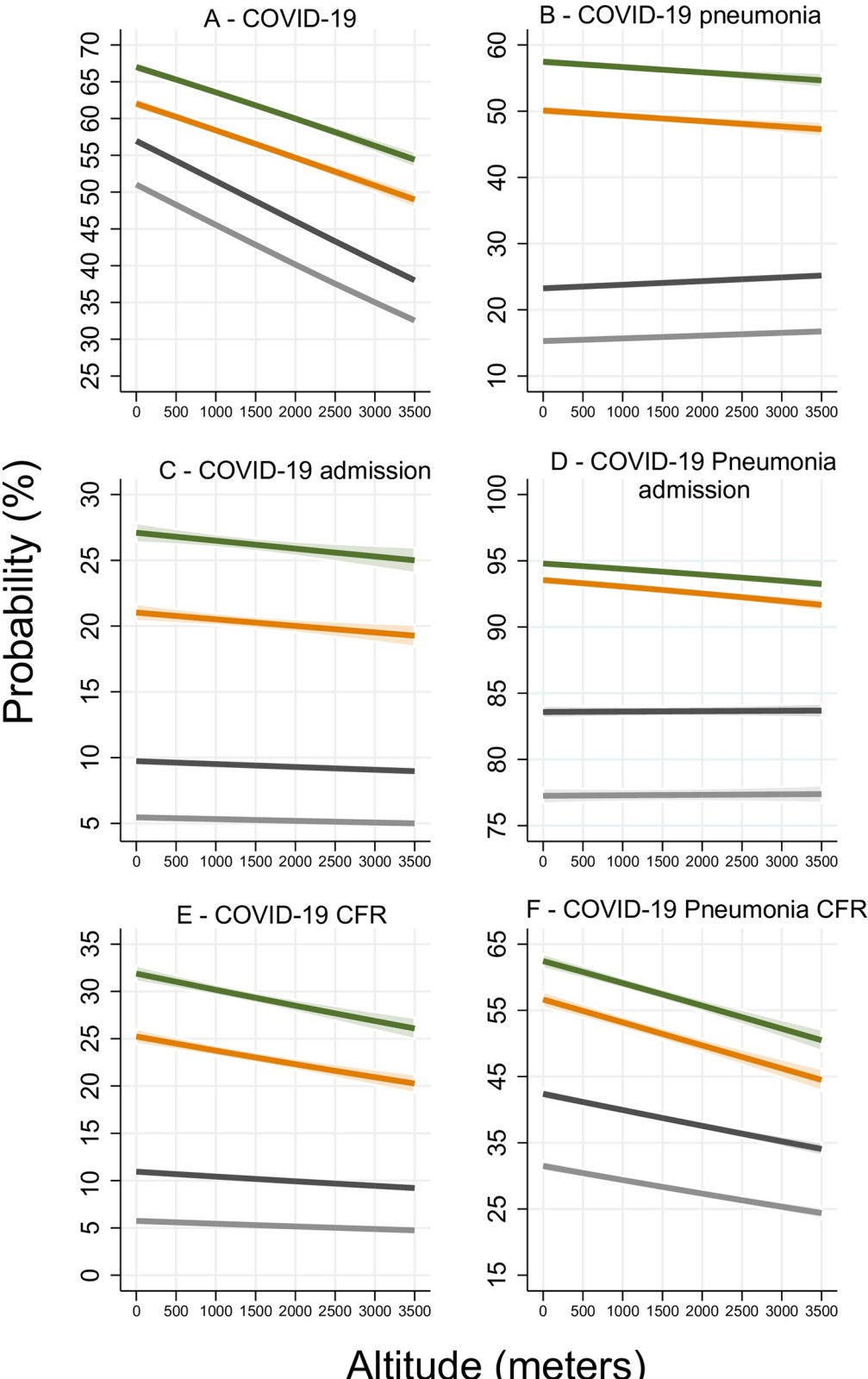

Green: DM, males Orange: DM, females. Dark gray: No DM, males. Light gray: No DM, females

**Fig 2. Event probability by altitude of residence.** Panel A-B show that the probability of COVID-19 and COVID-19 pneumonia in patients with DM decreased with higher altitudes. Panel C-D show that the probability of admission for COVID-19 and COVID-19 pneumonia decreased with higher altitudes for patients with DM and increased for patients without DM. Panel E-F shows that case fatality rate decreased with higher altitudes. All results are statistically significant at p<0.05.

glucose levels in pulmonary fluids that may favor viral replication, and impaired neutrophil activation because of uncontrolled hyperglycemic episodes that augment the risk of bacterial superinfection [26].

Several suggested mechanisms explain the clinical outcomes of patients with COVID-19 and DM, such as deregulation of the RAAS, impairment of the inflammatory response and hypercoagulable state, and the physiological and structural lung abnormalities caused by hyperglycemia [1, 27]. From these mechanisms, RAAS has a central role in SARS-CoV-2 infection and the related inflammatory response. This virus enters the host through the spike glyco-protein S, which binds to the angiotensin-converting enzyme 2 [ACE2] in pulmonary tissue. This enzyme is an important regulator of the RAAS since it leads anti-inflammatory and anti-fibrotic effects through the conversion of Ang II to Ang 1–7 [1]. When the virus binds to the host cell, it is internalized along with its receptor, resulting in a drastic decrease of cellular and soluble ACE2, which produces deregulation of the RAAS in favor of pro-inflammatory effects and increase of vascular permeability [28]. In this line, animal studies of induced DM showed

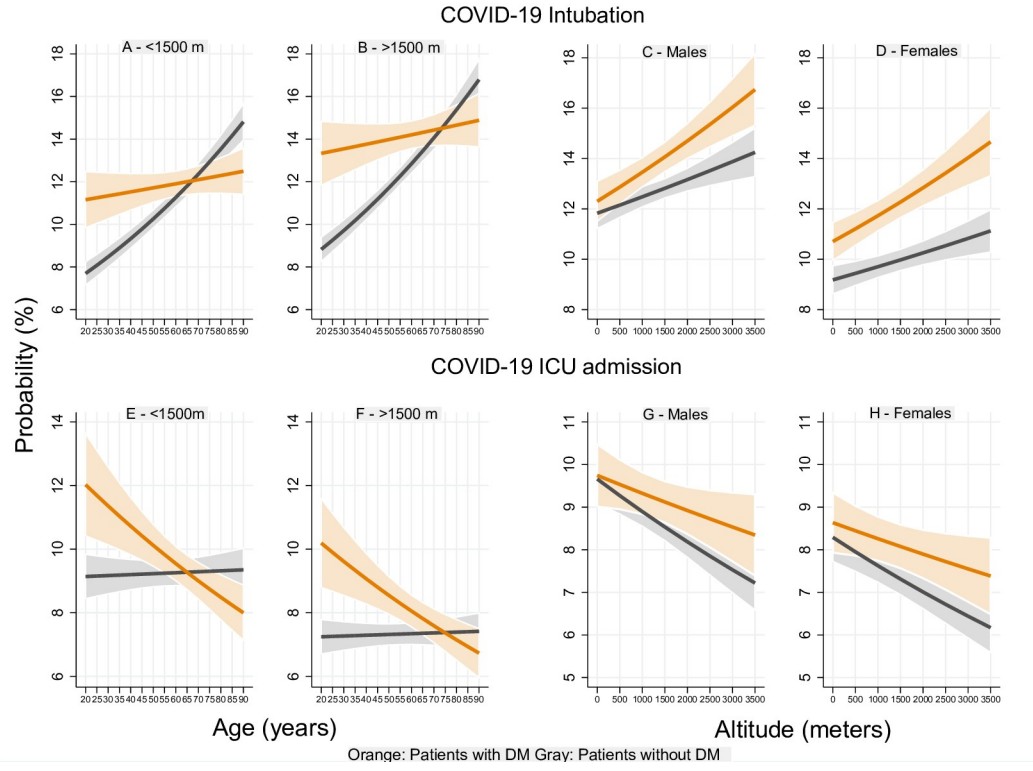

**Fig 3. Event probability by age and altitude of residence in patients with COVID-19.** Panel A-D show that the probability of intubation increased with age and altitude. This probability was higher for male patients [p<0.05] and for patients with DM up to 60 [at <1500 m] [p<0.05] and 65 years old [at >1500 m] [p<0.05]. Panel E-H show that the probability of ICU admission decreased with age for patients with DM and increased for patients without DM. This probability decreased with higher altitudes. No significant differences were found between men and women with or without DM at all altitudes [p≥0.05].

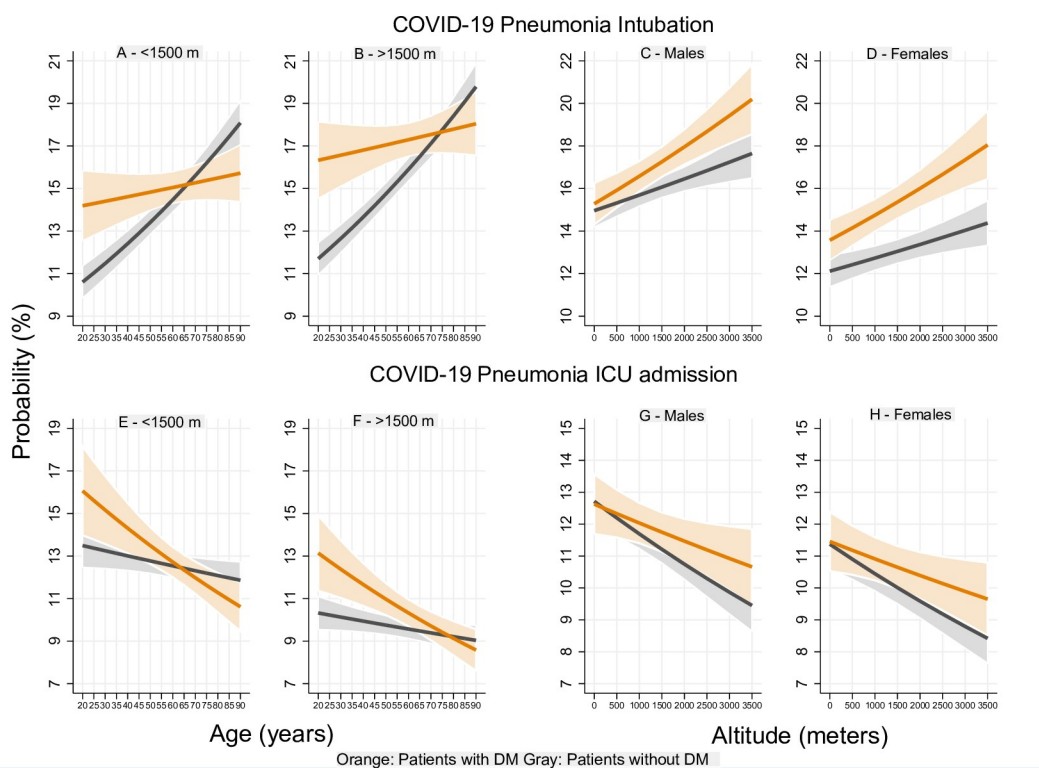

**Fig 4. Event probability by age and altitude of residence in patients with COVID-19 pneumonia.** Panel A-B show that the probability of intubation was higher for patients with DM up to 50 [at <1500 m] [p<0.05] and 65 years [at >1500 m] [p<0.05]. Panel C-D show that this probability was higher for males compared to that of women [p<0.05]. There were no differences between male patients with and without DM at all altitudes [p≥0.05]. Panel E-F show that there were no significant differences in ICU admission between patients with and without DM [p≥0.05]. Panel G-H show that this probability decreased with altitude and that there were no significant differences between men and women with or without DM [p≥0.05].

an increase in ACE2 levels in the lungs, heart, pancreas and kidneys [29]. In addition, a mendelian randomization study found that DM is causally associated with increased ACE2 expression in the lungs [30]. While this increase helps to counteract the metabolic inflammation produced by DM, it could also enhance the union of the virus to pulmonary tissue and shift the RAAS towards the inflammatory and fibrotic pathway [31]. DM also increases furin, another important enzyme involved in SARS-CoV-2 infection. This cellular protease facilitates viral entrance by cleavage of S1 and S2 domain of S protein to allow its union to the virus and promote infection [29].

Regulation of the immune system in COVID-19 is essential to prevent severe disease. Patients with DM have lost their capacity to regulate immunity [32], which makes them more susceptible to SARS-CoV-2 induced hyper-inflammation and cytokine storm [29]. DM is associated with impaired phagocytic activity of macrophages and decreased neutrophil chemotaxis. In addition, this metabolic disease produces inhibition of lymphocyte proliferation [27], which worsens lymphocytopenia caused by destruction of SARS-CoV-2 infected lymphocytes [33]. Guo et al. [15] reported that, compared to patients without DM, patients with DM and without other comorbidities have a higher risk of developing an excessive inflammatory response and significantly higher levels of inflammatory biomarkers such as IL-6, reactive C protein and serum ferritin. These results also suggest that patients with DM are more susceptible to develop a viral induced cytokine storm. In the same study, patients with DM had an increased

risk of developing a hypercoagulable state associated with significantly higher levels of coagulation markers such as D-dimer. Both insulin resistance and DM are associated with endothelial dysfunction and increased platelet activation and aggregation. As such, there is an imbalance between clotting factors and inhibition of the fibrinolytic system, which favors the development of a hypercoagulable state [27].

Regarding blood glucose level, SARS-CoV-2 infected patients have displayed poor glycemic control. A possible explanation could be that the increase in ACE2 glycosylation induced by hyperglycemia, allows viral binding to the receptor. Additionally, the cytokine storm generated by the viral infection, also affects glycemic control [5]. In other coronavirus family viruses' studies, development of hyperglycemia was observed in patients with SARS-CoV-1 without pre-existing DM. In these patients, hyperglycemia persisted for up to 3 years after recovery from SARS, indicating long-term damage to beta cells [34]. Furthermore, it is important to consider the effect of hyperglycemia in lung structure, function and outcome. DM is associated with thickening of the pulmonary basal lamina and the alveolar epithelium, along with reduced diffusing capacity for carbon monoxide [32].

Concerning the relationship between DM and high altitude, studies in Krygystan, Tibet, Chile, USA and Peru, have shown that the frequency of this disease decreases with increasing altitude [35–39]. This phenomenon was first reported in 1977 by Singh et al. [40], which compared 20,000 men stationed between 3692 to 5538 m with 130,700 men between 0 to 760 m. Similarly, Solis et al. [41] found a lower prevalence of DM in hospitalized patients at high altitude. At low altitude more women (p = 0.003) and men (p = 0.0002) had parents with DM, and more women had grandparents (p<0.0001) with DM than at high altitude [35].

Low risk of DM in high altitude populations may be explained by the low prevalence of obesity, overweight, sedentary lifestyle, and family history, compared to low altitude settings. Studies in high altitude residents have also registered lower blood glucose levels, obesity and DM [35]. A higher prevalence of obesity among females (p = 0.0005) and males (p<0.001) that live in low altitude settings was reported, compared to those living at high altitude. Similarly, residents of high-altitude settings in the Kyrgyz Republic had a lower prevalence of obesity, DM, cardiovascular risk factors and smoking, compared with low altitude residents [35, 42]. Nevertheless, the lower prevalence of self-reported DM at high altitude cannot be solely attributed to a lower prevalence of obesity [38, 43]. This premise is supported by a reported trend for lower odds of having DM among lean subjects living at high altitude compared with lean subjects living below 500 m [38].

Other studies that evaluated parameters closely related to DM found that individuals living at high altitude reported lower fasting glycemia levels [44–46] and better glucose tolerance [47–49] compared to individuals living at low altitudes. Additionally, studies with rodent and human skeletal muscle incubated under anoxic and hypoxic conditions registered an increased glucose uptake [49–51] and an increased content of the glucose transporter GLUT-4 [49–53].

Concerning the impact of altitude on COVID-19, frequency of infection decreased in high altitude populations of Tibet, Bolivia, Ecuador, Peru, Colombia, Ecuador, Brasil, USA, and other countries of the American continent [6, 7, 9, 54–61]. However, full lockdown in Peru caused an internal migration that through a specific land route was a significant factor progressively overriding the protection from COVID-19 afforded by high altitude [56].

Altitude exerts a protective effect for the development of diabetes, hypertension and obesity, comorbidities that may worsen the outcome of COVID-19 [62]. Also, the size of the virus inoculum in the air should gradually decrease as the barometric pressure decreases and the distance among air molecules increases [9]. Furthermore, expression of pulmonary ACE2 appears to decrease under physiological conditions of chronic hypoxia such as those found in high-altitude settings [6, 10, 11]. A study from Zhang et al. showed that under chronic hypoxia [2% $O_2$

for 12 days], ACE was upregulated by hypoxia inducible factor 1 [HIF-1] in human pulmonary artery smooth muscle cells, while the expression of ACE2 was markedly decreased [10]. Similar results obtained from studies with rats exposed to conditions equivalent to 4500 m, showed after 28 days increased levels of ACE and decreased expression of ACE2 in heart cells [11].

Hypobaric hypoxia appears to be a promising way to improve immune responses, decrease inflammation, and increase toleration of the "silent hypoxemia" state presented in COVID-19 [63]. Mice inoculated with influenza virus had lower viral load and survived more at simulated high altitude conditions [64]. However, attributing the lower frequency of COVID-19 in altitude only to hypobaric hypoxia is inappropriate, since in Peru the frequency of patients with influenza virus disease is only 30% lower at high altitude [65], while with SARS-CoV-2 is 350% lower [54].

In this line, Lhasa (the capital of Tibet at 3500 m) has been less affected by the pandemic compared to the rest of China. Similar observations were made in cities of Bolivia and Ecuador located over 2500 m, which present lower COVID-19 rates compared to cities located at sea level [6]. Recent data from 185 Peruvian capital provinces with altitudes ranging from 3 to 4342 m confirmed previous reports that SARS-CoV-2 infection is reduced at high altitudes [7]. In this line, the incidence and severity of COVID-19 together with the transmission capacity of the SARS-CoV-2 virus in the countries of the American continent decreases significantly from 1000 meters of altitude. In this work, all the data were normalized by population density [6]. These results were confirmed in a subsequent study [9].

Certain characteristics of high-altitude settings, such as less social interaction and greater distance between houses, must be considered as possible alternative explanations for lower COVID-19 rates [6]. In Brazil, population density is a highly associated factor of COVID-19 infection and death rates [59]. Although less population concentration is found at high altitude, in Peru altitude shows the same protective effect when normalized by million people or by population density (population/Km$^2$) [56]. Additionally, the lower infection and mortality rates at high altitude may be due to reduced access to health care services [57]. Furthermore, with every 1000 m, UV radiation levels increase by approximately 10%. UV radiation is a known germicidal that inactivates bacterial spores and mycobacteria [66], and the UV susceptibility coronavirus aerosols has been registered as 7 to 10 times that of other viral aerosols [67]. Thus, data collected to this date suggest that ambient levels of both UVA and UVB may be beneficial for reducing the severity and mortality of COVID-19 [68]. Other conditions, such as temperature and relative humidity, were inversely related to daily new cases and deaths. From data collected of 166 countries 1°C increase in temperature was associated with a 3.08% (95% CI: 1.53%, 4.63%) reduction in daily new cases and a 1.19% (95% CI: 0.44%, 1.95%) reduction in daily new deaths, whereas a 1% increase in relative humidity was associated with a 0.85% (95% CI: 0.51%, 1.19%) reduction in daily new cases and a 0.51% (95% CI: 0.34%, 0.67%) reduction in daily new deaths [69].

However, an ecological study that found that altitude is negatively associated with changes in COVID-19 prevalence and CFR, determined that population density only influenced CFR and not COVID-19 prevalence [8]. Therefore, ACE2 high altitude physiopathology could explain our results since the data collected demonstrated that the probability of COVID-19 was higher for patients who lived below 1500 m compared to that of those who lived above 1500 m (**Fig 1**). Overall, this probability decreased with higher altitudes (**Fig 2**). Additionally, we also demonstrated that for patients with DM, the probability of hospitalization (**Fig 2**), ICU admission (**Fig 3**), and the CFR (**Fig 2**) decreased with higher altitudes. Patients without DM shared the same results except for their probability of hospitalization that contrastively increased at higher altitudes (**Fig 2**).

The extensively explored dynamics between glucose homeostasis with exposure to altitude could explain the differences between patients with DM and patients without DM. So far, the mechanisms behind the apparent benefits of high altitude are not fully understood. Small clinical studies have shown that hypoxic sessions may improve glucose homeostasis in patients with DM. However, these effects must be explored further and over longer periods of time [70, 71].

Limitations of the study include lack of outpatient and inpatient laboratory and clinical data, treatment status of disease, socioeconomic variables that affect determinants of chronic diseases, and climatological variables that may influence COVID-19 spread. Socioeconomic and climatological variables were partially managed by regressing the individual patient data based on the current city of residence. Also, due to the nature of the database used, limitations as of data quality, completion, availability, granularity and the extent of the available variables at the individual level included should be considered.

## Conclusions

The study suggests that subjects with DM have a higher probability of being a confirmed COVID-19 and developing pneumonia. High altitude had a protective relationship with SARS-CoV-2 infection; however, it may be associated with more severe cases in patients with and without DM. High altitude decreases mortality for all COVID-19 patients. Regarding differences related to sex, our study shows that, compared to men, women with or without DM have a lower probability of COVID-19 infection, development of COVID-19 pneumonia, hospitalization, intubation, and admission to the ICU.

## Acknowledgments

We would like to extend our gratitude to the Secretariat of Health of Mexico for providing a free and open database of COVID-19 adult Mexican patients. Additionally, we are thankful to Ms. Teresa Roncal, International Consultant at the Urban, Resilience and Land Global Unit of the World Bank [Washington D.C.] for editing and proofreading the English version of the article. Ms. Roncal has not received any economic compensation and has accepted to have her name included in this section.

## Author Contributions

**Conceptualization:** Juan Alonso Leon-Abarca, Roberto Alfonso Accinelli.

**Data curation:** Juan Alonso Leon-Abarca.

**Formal analysis:** Juan Alonso Leon-Abarca.

**Investigation:** Juan Alonso Leon-Abarca, Arianna Portmann-Baracco, Mayte Bryce-Alberti, Carlos Ruiz-Sánchez.

**Methodology:** Juan Alonso Leon-Abarca, Roberto Alfonso Accinelli.

**Project administration:** Roberto Alfonso Accinelli, Jorge Soliz, Gustavo Francisco Gonzales.

**Resources:** Arianna Portmann-Baracco, Mayte Bryce-Alberti, Carlos Ruiz-Sánchez.

**Software:** Juan Alonso Leon-Abarca, Roberto Alfonso Accinelli.

**Supervision:** Roberto Alfonso Accinelli, Jorge Soliz.

**Validation:** Roberto Alfonso Accinelli, Jorge Soliz, Gustavo Francisco Gonzales.

**Visualization:** Juan Alonso Leon-Abarca.

**Writing – original draft:** Juan Alonso Leon-Abarca, Arianna Portmann-Baracco, Mayte Bryce-Alberti, Carlos Ruiz-Sánchez, Roberto Alfonso Accinelli, Gustavo Francisco Gonzales.

**Writing – review & editing:** Juan Alonso Leon-Abarca, Roberto Alfonso Accinelli, Jorge Soliz, Gustavo Francisco Gonzales.

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
