## [Decision Letter · Decision Letter 0]

17 May 2021

PONE-D-20-38743

Diabetes increases the risk of COVID-19 in an altitude dependent manner: an analysis of 1,280,806 Mexican patients

PLOS ONE

Dear Dr. Accinelli,

Thank you for submitting your manuscript to PLOS ONE. After careful consideration, we feel that it has merit but does not fully meet PLOS ONE’s publication criteria as it currently stands. Therefore, we invite you to submit a revised version of the manuscript that addresses the points raised during the review process.

The present investigation addresses an interesting topic, the relationship between DM and altitude in COVID disease. We suggest to shorten the discussion section and to focus on this peculiar relationship, trying to elucidate potential mechanisms. 

We look forward to receiving your revised manuscript.

Kind regards,

Chiara Lazzeri

Academic Editor

PLOS ONE

Journal Requirements:

2. In your Methods section, please provide additional information about the data collection methods and the demographic details of your participants. Please ensure you have provided sufficient details to replicate the analyses such as:

a) the dates the medical records were originally collected,

b) the dates on which you accessed the databases

c) a table of relevant demographic details,

d) the institutions that participants were treated at originally.

4. Please include a copy of Tables 1 and 2 which you refer to in your text on page 4 and 5.

Reviewers' comments:

Reviewer's Responses to Questions

**Comments to the Author**

1. Is the manuscript technically sound, and do the data support the conclusions?

Reviewer #1: Yes

2. Has the statistical analysis been performed appropriately and rigorously? 

Reviewer #1: Yes

3. Have the authors made all data underlying the findings in their manuscript fully available?

Reviewer #1: Yes

4. Is the manuscript presented in an intelligible fashion and written in standard English?

Reviewer #1: Yes

5. Review Comments to the Author

Reviewer #1: p3 ln 92 - constraints of this study = better suited for the discussion section

p3 ln 100 - ethics etc - name of the committee that Ok-ed the study and date of approval

Discussion - Less on DM and Covid-19 (which is widely known) - focus on attitude and Covid-19 (which is less well known)

6. PLOS authors have the option to publish the peer review history of their article (what does this mean?). If published, this will include your full peer review and any attached files.

Reviewer #1: **Yes: **Ioannis Ilias

---

## [Author Response · Author response to Decision Letter 0]

6 Jul 2021

a) The style has been corrected

2. In your Methods section, please provide additional information about the data collection methods and the demographic details of your participants. Please ensure you have provided sufficient details to replicate the analyses such as:

a) the dates the medical records were originally collected,

i. Data was accesed at Sept 9th 2020. As the Mexican government releases their databases to the public domain with a 24 hour delay, the available registers dated until Sept 8th 2020.

b) the dates on which you accessed the databases

i. Now specified in methods: Data was accesed at Sept 9th 2020.

c) a table of relevant demographic details

i. Now available as “Table 1”

d) the institutions that participants were treated at originally.

i. As the list of healthcare institutions is extensive due to the nature of the database (nation-wide), we specifify in “Methods” that we included all patients treated along the 475 USMERs (Monitoring Units of Viral Respiratory Infections, acronym in spanish)

a) The style has been corrected

4. Please include a copy of Tables 1 and 2 which you refer to in your text on page 4 and 5.

a) Tables 1 and 2 are now attached to the manuscript

5. Reviewer #1: p3 ln 92 - constraints of this study = better suited for the discussion section

a) Constraints were added to the discussion section.

6. p3 ln 100 - ethics etc - name of the committee that Ok-ed the study and date of approval

a) We wrote: “approval from an institutional ethics committee”

b) Now it says: “The study received approval from Universidad Peruana Cayetano Heredia’s ethics committee with registration number SIDISI 202908. “

7. Discussion - Less on DM and Covid-19 (which is widely known) - focus on attitude and Covid-19 (which is less well known)

a) The discussion now focuses on the possible pathophysiology related to our findings.

---

## [Editor Report · Decision Letter 1]

12 Jul 2021

Diabetes increases the risk of COVID-19 in an altitude dependent manner: an analysis of 1,280,806 Mexican patients

PONE-D-20-38743R1

Dear Dr. Accinelli,

We’re pleased to inform you that your manuscript has been judged scientifically suitable for publication and will be formally accepted for publication once it meets all outstanding technical requirements.

Kind regards,

Chiara Lazzeri

Academic Editor

PLOS ONE
---

## [Editor Report · Acceptance letter]

21 Jul 2021

PONE-D-20-38743R1 

Diabetes increases the risk of COVID-19 in an altitude dependent manner: an analysis of 1,280,806 Mexican patients. 

Dear Dr. Accinelli:

I'm pleased to inform you that your manuscript has been deemed suitable for publication in PLOS ONE. Congratulations! Your manuscript is now with our production department. 

Kind regards, 

on behalf of

Dr. Chiara Lazzeri 

Academic Editor

PLOS ONE